# Evaluation of Fermented Extracts of *Aloe vera* Processing Byproducts as Potential Functional Ingredients

**Seong-Hun Lee [1,†], Chang-Ho Eun [2,†], Yong-Seong Kwon [1], Jin-Hong Baek [1] and In-Jung Kim [3,4,*]**

[1] KJM Aloe R&D Center, Seoqwipo 63636, Korea; showbii7@gmail.com (S.-H.L.); yskwon@aloe.co.kr (Y.-S.K.); backga@aloe.co.kr (J.-H.B.)

[2] Subtropical Horticulture Research Institute, Jeju National University, Jeju-si 63243, Korea; mong6908@gmail.com

[3] Faculty of Biotechnology, College of Applied Life Science, Jeju National University, Jeju-si 63243, Korea

[4] Bio-Resources Computing Research Center, Jeju National University, Jeju-si 63243, Korea

\* Correspondence: ijkim@jejunu.ac.kr; Tel.: +82-64-754-3357

† Seong-Hun Lee and Chang-Ho Eun are contributed equally to this work.

**Abstract:** Aloe is widely used as a cosmetic and medicinal plant. Numerous studies have reported that aloe gel extract has antioxidant, anticancer, antidiabetic, immunity, and skin antiaging properties. However, few studies have investigated the properties of fermentation products of aloe processing byproducts. Aloe stalks and leaves remain as byproducts after the aloe beverage manufacturing process. This study evaluated whether fermentation products of blender and press extracts of aloe processing byproducts (BF and PF, respectively) that remain after beverage manufacturing were useful as functional biomaterial by investigating their effects on adipocyte differentiation, hyaluronic acid (HA) production, tyrosinase activity, and antioxidant activity. Co-fermentation of *G. xylinus* and *S. cerevisiae* was conducted for fermentation of aloe processing byproducts. The BF and PF products did not induce observable cytotoxicity effects. However, BF and PF products did inhibit a 3T3-L1 adipocyte differentiation compared with control, with the BF product displaying greater inhibition of 3T3-L1 adipocyte differentiation than the PF product. HA production increased in HaCaT cell cultures as the concentration of the MF product increased, as compared with the untreated control. The levels of tyrosinase inhibition, 2,2-diphenyl-1-picrylhydrazyl (DPPH) radical scavenging, and superoxide dismutase (SOD)-like activity also depended on the MF product concentration. This study indicates that the fermented products of aloe processing byproducts have biological potential for applications in the manufacturing of cosmetics, pharmaceuticals, and beverages. These laboratory bench results provide the foundation for future studies of scaling and practical applications at the industrial level.

**Keywords:** adipocyte differentiation inhibition; aloe byproduct; antioxidant; fermentation; hyaluronic acid; tyrosinase inhibition

## 1. Introduction

Aloe vera (*Aloe barbadensis* Mill.) is a perennial herb belonging to the Aloineae tribe of the Liliaceae family and is primarily cultivated in the tropics [1–3]. More than 400 species are known worldwide, but currently, only 6–7 species are used for food and medicinal purposes. *Aloe vera* and *Avore sense* are the primary species cultivated in Korea. *Aloe vera* leaves consist of more than 90% water, can be divided into a green peel on the outside and a gel on the inside, and contain approximately 300 metabolites, including polysaccharides, vitamins, fatty acids, amino acids, selenium, calcium, and anthraquinone. Aloe has long been used as a medicinal plant because of its beneficial effects on gastric ulcers, skin irritation, burns, cancers, immune activity, and skin diseases [4–9]. The use of aloe products in functional foods is expanding, including immune-boosting foods and health

supplements containing polysaccharides, flavonoids, anthraquinone, gibberellin, and beta-sitosterol [10–12]. On the other hand, toxicity and adverse clinical effects of aloe extracts have also been reported [13,14].

The natural food industry generates a large amount of waste and processing byproducts. Disposal of these products causes financial loss and legal oversight. As the amount of waste produced during food manufacturing increases, it becomes a potential cause of environmental pollution. Continuous efforts to separate and utilize useful biomaterials from these wastes and byproducts could resolve the associated economic, legal, and environmental issues. These efforts are considered a promising R&D field that is consistent with the Green New Deal policy in Korea [15,16].

Most studies of the functional effects of aloe focus on the antioxidant, anticancer, antidiabetic, immune enhancement, and whitening effects of aloe gel extract, and several studies investigated the use of aloe gel fermentation products. Hai et al. reported that *Aloe vera* fermentation products are effective in healing burn injuries by reducing the severity of inflammation [17]. Jiang et al. showed that aloe fermentation supernatant containing *Lactobacillus plantarum* protects human intestinal health, delays senescence, and prevents chronic diseases [18]. Al-Madboly et al. reported that butyrate fermented by endophytic microbiota in *Aloe vera* gel affects inflammatory responses by reducing the levels of reactive oxygen species [19]. However, few studies have investigated the properties of fermentation products of aloe processing byproducts remaining after the aloe beverage manufacturing process. *G. xylinus* and *S. cerevisiae* used in this study are representative microorganisms widely used for fermentation products [20–22]. The production of antimicrobials and lactic acid using the fermentation products of processing byproducts of other plants such as tomato, carrot, melon, and orange have been reported [23,24].

With the aim of making and evaluating beverage and mask pack prototypes, this study investigated the effects of the fermented products of *Aloe vera* processing byproducts, such as stalks and leaves, remaining after beverage manufacturing on the inhibition of adipocyte differentiation, moisturizing, whitening, and antioxidant activity. The results of this study indicate that these fermented products could be utilized as functional biological materials for cosmetic, pharmaceutical, and beverage manufacturing.

## 2. Materials and Methods

### 2.1. Plant Material

The aloe stalks and whole leaves that remained after aloe beverage manufacturing were weighed at a ratio of 50:50. This study used the extract of these *Aloe vera* processing byproducts.

### 2.2. Cells and Reagents

The human keratinocyte cell line (HaCaT) was purchased from Cell Lines Service (CLS, Eppelheim, Germany). The preadipocyte cell line (3T3-L1) was purchased from the Korean Cell Line Bank (KCLB, Seoul, Korea). *Streptococcus mutans* (ATCC 25175) and *Porphyromonas gingivalis* (ATCC 33277) were purchased from the Korean Collection for Type Cultures (KCTC, Jeongeup, Korea). *Gluconacetobacter xylinus* subsp. *xylinus* (KCCM 41431) was purchased from the Korean Culture Center of Microorganisms (KCCM, Seoul, Korea). *Saccharomyces cerevisiae* (KACC 30008) was obtained from the Korean Agricultural Culture Collection (KACC, Jeonju, Korea). The EZ-Cytox Cell Viability Assay kit was purchased from Sigma-Aldrich (St. Louis, MO, USA), 3-[4,5-dimethylthiazol-2-yl]-2,5 diphenyl tetrazolium bromide (MTT), isopropanol, fetal bovine serum [FBS], 3-isobutyl-1-methylxanthine, dexamethasone, insulin, paraformaldehyde, Oil Red O, retinoic acid, phosphate buffer solution, L-DOPA, mushroom tyrosinase, kojic acid, 2,2-diphenyl-1-picrylhydrazyl (DPPH), ascorbic acid, and pyrogallol were purchased from Merck (Seoul, Korea). Dulbecco's modified eagle medium (DMEM), calf serum, and penicillin-streptomycin were purchased from Thermo Fisher (Seoul, Korea). Tris-HCl (pH 8.5) was purchased from Bioneer (Seoul, Korea).

### 2.3. Extraction of byproducts after Aloe vera Processing

*Aloe vera* processing byproducts were extracted using a blender and a press. For blender extraction, the byproduct was pulverized with a food blender (SAM WOO, SF-600, Seoul, Korea). For press extraction, the byproduct was pressed with a juicer (Hurom, HVS-STF14, Seoul, Korea). Then, the extract was heated at 90 °C for 20–30 min, centrifuged at $1000\times g$ rpm for 15 min, and filtered (HYUNDAI, HM.3005090, Seoul, Korea) to remove solids.

### 2.4. Fermentation Conditions for Aloe byproduct Extracts

The composition medium was prepared by mixing aloe byproduct extract, black tea, and 10% sugar water. Mass fermentation was conducted using a tray with a large surface area (27 $\times$ 40 $\times$ 7 cm). The blender or press extract of the aloe processing byproduct (600 mL), 10% sugar water (800 mL), bacterial culture (200 mL), and black tea (400 mL) were added to the tray, and the mixture was fermented at 25 °C for 20 days. The bacterial culture contained $1 \times 10^8$ cfu/mL of *G. xylinus* subsp. *xylinus* and $1 \times 10^8$ cfu/mL *S. cerevisiae* at a ratio of 8:2. The fermentation products of the blender and press extracts of aloe processing byproducts are hereafter referred to as BF and PF, respectively (Table 1). The experiments testing tyrosinase inhibition activity, 2,2-diphenyl-1-picrylhydrazyl (DPPH) radical scavenging activity, and superoxide dismutase (SOD)-like activity were performed using freeze-dried samples of fermentation products.

**Table 1.** Mass fermentation conditions of the aloe byproduct extract.

| | Fermentation Volume (2L) | | | | |
|---|---|---|---|---|---|
| | Aloe Extract (mL) | Bacterial Culture (mL) | Tea with 10% Sugar (mL) | Tea (mL) | Water (mL) |
| CF | 0 | 200 | 800 | 400 | 600 |
| BF | 600 (blender extract) | 200 | 800 | 400 | 0 |
| PF | 600 (press extract) | 200 | 800 | 400 | 0 |

CF: fermentation product without extract of the aloe byproduct; BF: fermentation product from the blender extract of the aloe byproduct; PF: fermentation product from the press extract of the aloe byproduct.

### 2.5. MTT Assay of Cell Proliferation

Undifferentiated 3T3-L1 cells were seeded in a 96-well cell culture plate at a density of $1 \times 10^4$ cells/well (100 μL) and cultured for 24 h at 37 °C and 5% $CO_2$ conditions. The cells were observed under a microscope to confirm that they grew and distributed on the bottom of the culture plate. Then, the medium was removed from each well, and 100 μL of medium was added containing the fermented products (at 1, 2, 4, 6, 8, and 10%). The cultures were incubated for 24 h at 37 °C and 5% $CO_2$ conditions. After incubation with the fermentation products, the medium was removed from each well, and 50 μL of 3-[4,5-dimethylthiazol-2-yl]-2,5 diphenyl tetrazolium bromide (MTT) reagent (1 mg/mL) was added to each well. The plate was covered with foil to block light and incubated at 37 °C for 2 h. After confirming that the cells contained a purple precipitate, the MTT reagent was removed from each well, and 100 μL of isopropanol was added to the wells. A microplate reader (Thermo Scientific, Singapore) was used to measure the reaction absorbance (at 570 nm) and the reference absorbance (at 650 nm).

### 2.6. Oil Red O Staining of Neutral Lipids

The 3T3-L1 adipocytes were cultured at 37 °C under 5% $CO_2$ conditions. The medium was replaced every 2–3 days with fresh Dulbecco's modified eagle medium (DMEM) containing 10% calf serum and 100 unit/mL penicillin-streptomycin. Two days after the cells became confluent, the medium was replaced with a cell differentiation induction medium (DMEM containing 10% fetal bovine serum [FBS], 100 unit/mL penicillin-streptomycin,

500 μM 3-isobutyl-1-methylxanthine, 0.25 μM dexamethasone, and 10 μg/mL insulin) and 2% BF and PF were added. Two percent CF was used as control. Cell differentiation was terminated by replacing the medium and culturing the cells in DMEM containing 10% FBS and 10 μg/mL insulin for 2 days. Then, lipid accumulation was analyzed using the Oil Red O (ORO) staining method. Briefly, differentiated cells were washed with phosphate-buffered saline (PBS) and fixed in 4% paraformaldehyde at room temperature for 30 min. To prevent background staining, 60% isopropanol was added to the wells and incubated at room temperature for 5 min. After washing with distilled water, 100 μL of 100% isopropanol was added to each well to dissolve intracellular ORO. Absorbance was measured at 517 nm. Lipid accumulation was calculated as a percentage of the amount measured in control cells (differentiated adipocytes that were not treated with the fermented product).

### 2.7. WST Assay of Cell Proliferation

The viability of HaCaT cells was analyzed using the EZ-Cytox Cell Viability Assay kit. HaCaT cells were inoculated in a 24-well plate at a concentration of $1.0 \times 10^5$ cells/well and cultured for 24 h. at 37 °C and 5% $CO_2$ conditions. The medium was exchanged with a FBS-free medium containing different contents of fermentation products (0, 0.16, 0.31, 0.63, 1.25, 2.5, and 5%), and the cells were cultured for 24 h. Then, 50 μL of EZ-Cytox was added to each well, cells were cultured for 1 h, the culture medium was transferred to a new 96-well plate, and absorbance was measured at 450 nm using a microplate reader. The average absorbance value for each sample group was measured, and the cell viability was determined by comparing the absorbance value with that of the control group.

### 2.8. Analysis of Hyaluronic Acid Production

HaCaT cells were seeded in a 24-well plate at a density of $1.0 \times 10^5$ cells/well and cultured for 24 h. Then, the medium was replaced with a FBS-free medium containing different volumes of fermentation products (at 0, 0.16, 0.31, and 0.63%), and cells were cultured for 24 h. The amount of hyaluronic acid (HA) in the culture medium was determined using the human HA ELISA kit (CUSABIO TECHNOLOGY, LLC) and measuring absorbance at 450 nm. Retinoic acid was used as the positive control.

### 2.9. Measuring the Inhibition of Tyrosinase Activity

The inhibition of tyrosinase activity was evaluated by measuring the levels of dihydroxyphenylalanine (DOPA) product produced by tyrosinase activity during the conversion of L-tyrosine to melanin. The reaction mixture contained 100 μL of 0.5 M phosphate buffer solution (pH 6.5), 40 μL of 10 mM L-DOPA, 20 μL of fermented product (1, 2, 4 mg/mL, and 40 μL of mushroom tyrosinase (110 U/mL), and the reaction was incubated at 25 °C for 2 min. The absorbance of dopachrome produced in the reaction was measured at 475 nm. Kojic acid was used as a positive control.

### 2.10. Measuring DPPH Radical Scavenging Ability

Assays to measure DPPH radical scavenging activity were performed as described by Heo et al. [25]. 160 μL of 0.4 mM DPPH and 40 μL of sample were added to a 96-well plate and incubated at 37 °C in the dark for 30 min. Absorbance was measured at 515 nm using a microplate reader (VersaMax, Molecular Devices, San Jose, CA, USA). Ascorbic acid was used as a positive control. DPPH radical scavenging activity was calculated as follows: DPPH radical scavenging activity (%) = $(1 - A/B) \times 100$, where A is the absorbance of the sample treatment group, and B is the absorbance of the control group (distilled water).

### 2.11. Measuring SOD-like Activity

SOD-like activity was measured according to the method of Lee et al. [26]: 40 μL of fermented products, 120 μL of Tris-HCl (pH 8.5), and 40 μL of 7.2 mM pyrogallol were added to a 96-well plate and incubated at 25 °C for 10 min. Absorbance was measured at 420 nm using a microplate reader (VersaMax).

### 2.12. Statistical Analysis

The results reported in this study have been expressed as the mean $\pm$ SD of three replicates. All statistical analyses were carried out using IBM SPSS software (SPSS for Windows, version 20, SPSS Inc., Armonk, NY, USA). Significant differences among the samples were calculated using a one-way analysis of variance (ANOVA), followed by Tukey's test at the 5% level ($p < 0.05$).

## 3. Results and Discussion

### 3.1. Cytotoxicity of the Aloe byproduct Fermentation Products

The cytotoxicity of the CF, PF, and MF fermentation products was investigated by performing the MTT assay using undifferentiated 3T3-L1 cells (Figure 1). In the MTT assay, the potential cytotoxicity of the fermentation products is inversely related to cell viability. In our study, we considered the fermentation product as cytotoxic when the cell viability decreased to less than 80%. The 3T3-L1 cell cultures were treated with 0, 1, 2, 4, 6, 8, and 10% of CF, PF, and MF products to examine cytotoxicity. The control cells were treated with the fermentation product without the aloe byproduct extract (CF) (Figure 1A). The PF product exhibited cytotoxicity with the addition of 4% of PF extract (Figure 1B). By contrast, adding the MF product did not induce cytotoxicity (Figure 1C).

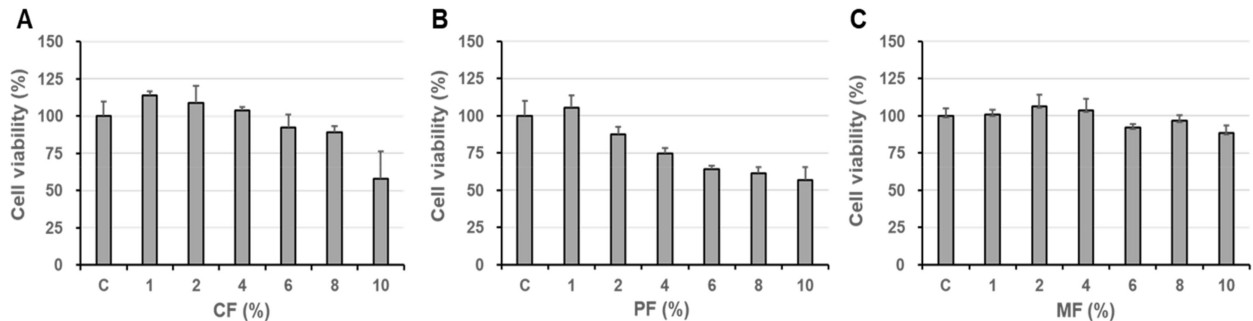

**Figure 1.** The viability of 3T3-L1 cells after the addition of different volumes of aloe byproduct fermentation products and extracts. (**A**) Control cells were treated with the fermentation product without the aloe byproduct extract (CF). (**B**) Cells treated with the PF extract of the aloe byproduct. (**C**) Cells treated with the MF extract of the aloe byproduct. CF, the fermentation product without the aloe byproduct extract; MF, fermentation product from the blender extract of the aloe byproduct; PF, fermentation product from the press extract of the aloe byproduct. Data are expressed as mean $\pm$ SD ($n$ = 3), and different small letters indicate a significant difference by Tukey's test; $p < 0.05$.

### 3.2. The Fermentation Product Inhibits 3T3-L1 Adipocyte Differentiation

Obesity is a global medical problem that causes several related diseases [27,28]. Adipocytes are the main component of white adipocyte tissue. An increase in preadipocytes leads to an increasing mass of adipocyte tissue in vivo [29]. The preadipocyte 3T3-L1 cell line is an in vitro model of adipogenesis and adipocyte differentiation [30]. Our study investigated whether fermentation products and extracts of aloe processing byproducts reduced body fat. The inhibitory effects of the fermentation product extracts on 3T3-L1 adipocyte differentiation were confirmed by ORO staining after the addition of 2% MF and PF extracts to 3T3-L1 cultures compared with the control (Figure 2A). The MF extract induced greater inhibition of 3T3-L1 differentiation (85.85 $\pm$ 0.14) than the PF extract (Figure 2B). Guo et al. (2020) reported that strawberry fermented with *Cordyceps militaris* has antiadipogenesis activity [31]. Other studies reported that fermentation extracts of *Curcuma longa* L. and soy germ prevent obesity by suppressing adipogenesis and promoting lipolysis [32,33]. Our results indicated that the MF product inhibited adipocyte differentiation by suppressing adipogenesis, and may have beneficial effects as a functional beverage to promote antiadipogenesis activity.

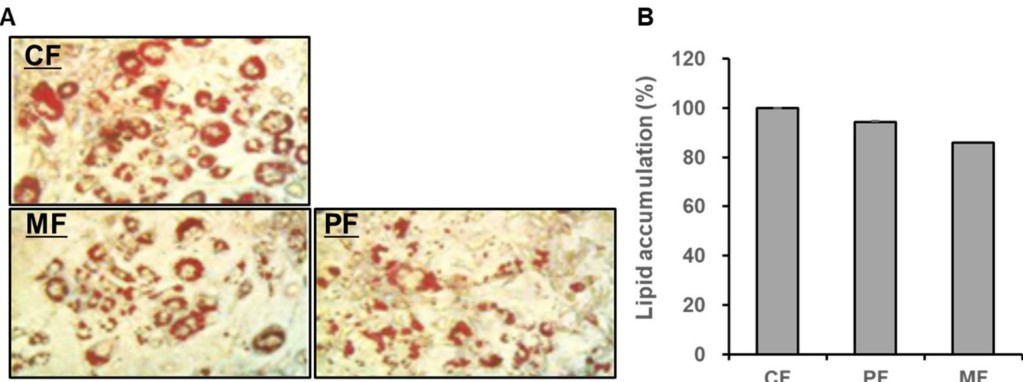

**Figure 2.** Effects of the fermentation product extracts (MF and PF) of the aloe byproducts on lipid accumulation during 3T3-L1 adipocyte differentiation. (**A**) Micrographs of Oil Red O-stained adipocytes. (**B**) Stained oil droplets were dissolved in isopropyl alcohol and quantified by reading the absorbance at 517 nm. CF, the fermentation product without the aloe byproduct extract; MF, fermentation product from the blender extract of the aloe byproduct; PF, fermentation product from the press extract of the aloe byproduct. Data are expressed as mean ± SD ($n$ = 3), and different small letters indicate a significant difference by Tukey's test; $p < 0.05$.

### 3.3. Effect of the Fermentation Product on Hyaluronic Acid Production

HA is an important component of the cellular matrix and has high moisture retention. Generally, HA can be extracted from various vertebrate tissues or produced by bacterial fermentation of *Streptococci* and *Pseudomonas aeruginosa* [34–36]. As the MF product strongly inhibited 3T3-L1 adipocyte differentiation, we investigated the effect of the MF product on HA production. First, we performed WST assays to determine whether the MF product displayed cytotoxicity to skin keratinocytes (HaCaT). We added the MF product to HaCaT cell cultures at 0.16%, 0.31%, 0.63%, 1.25%, 2.5%, and 5.0% volumes. The results showed that the 2.5% and 5.0% treatments were slightly cytotoxic compared with the control, whereas the remaining treatments (0.16%, 0.31%, 0.63%, and 1.25%) did not display cytotoxicity (Figure 3A). Based upon these results, HA production was evaluated after adding the MF products to HaCaT cell cultures at 0.16%, 0.31%, and 0.65% volumes. The results showed that HA production increased as the MF product concentration increased compared with the control (Figure 3B). This result indicates that the fermentation product of aloe processing byproducts can stimulate microbial HA production and may be effective in skin moisturizing applications. Amado et al. reported that HA is efficiently produced by microbial fermentation from agroindustrial byproducts [37]. Marine byproducts, such as mussel processing wastewater and tuna peptone viscera, can be used for the production of HA by *Streptococcus zooepidemicus* [38].

### 3.4. The Fermentation Product Inhibits Tyrosinase Activity

Skin exposure to UV rays causes tyrosine in melanocytes to oxidize through the biosynthesis of tyrosinase to produce melanin [39,40]. Excessive melanin can cause hyper-pigmentation and melasma. Inhibition of the tyrosinase enzyme that produces melanin confers a skin-whitening effect. We investigated whether the MF product inhibited tyrosinase activity by measuring the dihydroxyphenylalanine product produced by the tyrosinase enzyme. The results showed that the MF product inhibited tyrosinase activity by 17.06% ± 0.13% at 1 mg/mL, 43.61% ± 0.25% at 2 mg/mL, and 66.60% ± 0.38% at 4 mg/mL, thereby confirming that inhibition of tyrosinase activity depended on the MF product concentration (Figure 4). Rice bran, the byproduct of rice processing, contains functional metabolites and phenolic compounds, and the product of fungal fermentation inhibited tyrosinase and elastase activity [41,42]. Razak et al. reported that broken rice fragments generated during the milling process generated effective tyrosinase and elastase inhibition activities [43].

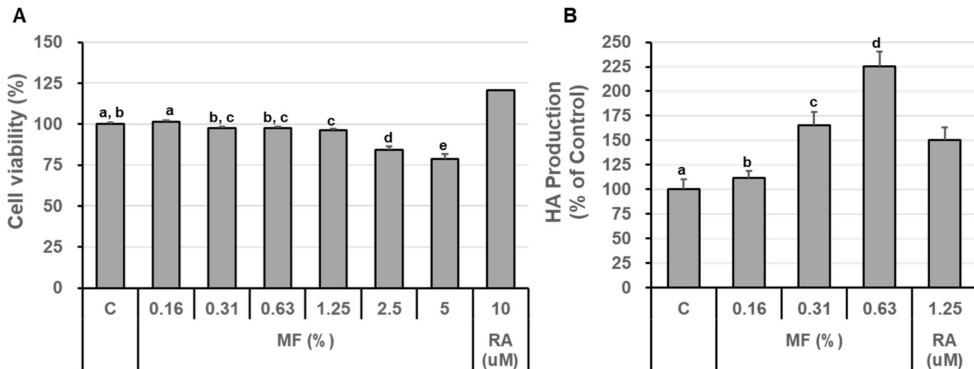

**Figure 3.** (**A**) The viability of HaCaT cells after the addition of different volumes of MF products. Cell cultures were treated with indicated volumes of the MF product for 24 h at 37 °C. Cell viability was measured by the WST assay. (**B**) Hyaluronic acid (HA) production in HaCaT cells treated with MF products. MF, fermentation product from the blender extract of the aloe byproduct; RA, retinoic acid. Data are expressed as % of control (mean ± SD, $p < 0.05$, MF product vs. control). C, control without MF. Data are expressed as mean ± SD ($n = 3$), and different small letters indicate a significant difference by Tukey's test; $p < 0.05$.

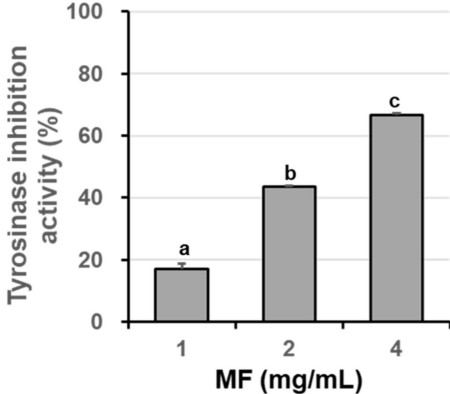

**Figure 4.** The MF extract of aloe processing byproducts inhibits tyrosinase activity. MF, fermentation product from the blender extract of the aloe byproduct. Data are expressed as mean ± SD ($n = 3$), and different small letters indicate a significant difference by Tukey's test; $p < 0.05$.

*3.5. The Fermentation Product Displays Antioxidant Activity*

DPPH radical is a stable free radical that transforms into a stable molecule by accepting electrons or protons from other atoms and molecules [44]. We analyzed the scavenging activity of different concentrations of the MF product against the DPPH radical. The DPPH radical scavenging activity of the MF product was 84.11% ± 0.03% at 1 mg/mL, 89.28% ± 0.65% at 2 mg/mL, and 92.01% ± 0.13% at 4 mg/mL, indicating that the MF product exhibited concentration-dependent radical scavenging activity (Figure 5A). The ascorbic acid standard at 50 µg/mL exhibited more than 90% radical scavenging activity (Figure S1). We also investigated whether the MF product inhibits oxidation in a similar manner as SOD. SOD-like activity was determined by measuring pyrogallol autoxidation, which reacts with superoxide to cause browning (Figure 5B). The SOD-like activity of the MF product was 15.10% ± 3.79% at 1 mg/mL, 41.60% ± 0.04% at 2 mg/mL, and 64.56% ± 0.04% at 4 mg/mL, indicating that the SOD-like activity depended on the MF product concentration. The ascorbic acid standard exhibited approximately 51% SOD-like activity at a concentration of 100 µg/mL (Figure S2). Antioxidant and prebiotic activities were enhanced in the solid fermented extract of the grape byproduct [45]. Recent reviews discuss several studies on antioxidants and natural preservatives isolated from food byproducts through fermentation [46,47].

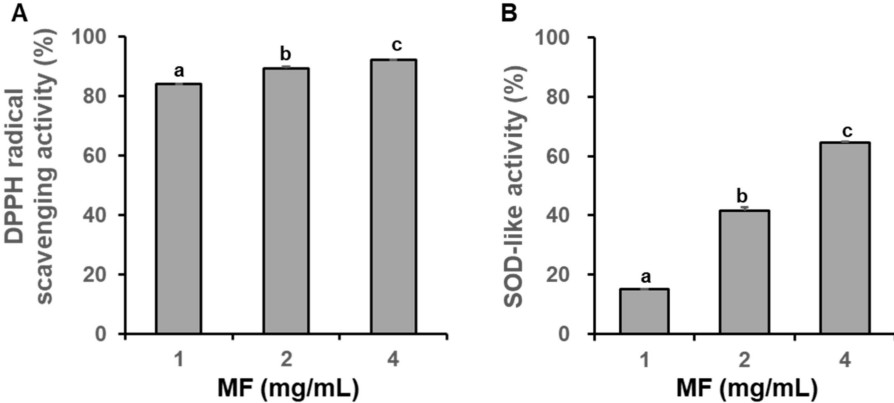

**Figure 5.** Antioxidant activity of the MF product of the aloe byproducts. (**A**) DPPH radical scavenging activity. (**B**) SOD-like activity. DPPH, 2,2-diphenyl-1-picrylhydrazyl; SOD, superoxide dismutase; MF, fermentation product from the blender extract of the aloe byproduct. Data are expressed as mean $\pm$ SD ($n = 3$), and different small letters indicate a significant difference by Tukey's test; $p < 0.05$.

## 4. Conclusions

The results of this study suggest that the aloe processing byproduct can be successfully fermented by a combination of *G. xylinus* and *S. cerevisiae*, and the fermented product of a blender extract of the aloe processing byproduct contains compounds that can be used for antiobesity, HA production, antityrosinase, and antioxidant applications. Therefore, this study demonstrates the biological potential of the fermented product of the aloe processing byproduct, which is expected to have applications in the cosmetic, pharmaceutical, and beverage industries. These results also suggest that other fermented products of food industry byproducts could have biological potential, and their use would reduce the environmental impact of food industry byproducts. However, more research on the scaling process is required to put these laboratory results into practical use at the industrial level. In Korea, cosmetics and related products are regulated by the Korea Food and Drug Administration (KFDA) before they are allowed in Korea. Therefore, in order to prove the efficacy and potential as an effective ingredient of an aloe-fermented product, it is necessary to evaluate additional biophysical parameters, such as moisturizing, depigment, skin whitening, or antiaging effects. In addition, the fermented products in this study need to be further characterized in terms of structure and composition, as well as biologically, in order to be accepted as functional ingredients or nutraceuticals of aloe beverages and mask packs.

**Supplementary Materials:** The following are available online at https://www.mdpi.com/article/10.3390/fermentation7040269/s1, Figure S1: Calibration curve of ascorbic acid for DPPH radical scavenging activity, Figure S2: Calibration curve of ascorbic acid for SOD-like activity.

**Author Contributions:** Conceptualization, I.-J.K.; Investigation, J.-H.B.; Project administration, S.-H.L.; Resources, Y.-S.K. and J.-H.B.; Supervision, C.-H.E. and I.-J.K.; Validation, Y.-S.K.; Writing—original draft, S.-H.L. and C.-H.E.; Writing—review and editing, I.-J.K. All authors have read and agreed to the published version of the manuscript.

**Funding:** This research was funded by the Basic Science Research Program through the National Research Foundation of Korea [2019R1A6A1A11052070] and the Ministry of SMEs and Startups [P0010755].

**Institutional Review Board Statement:** Not applicable.

**Informed Consent Statement:** Not applicable.

**Data Availability Statement:** The data presented in this study are available on request from the corresponding author.

**Acknowledgments:** We are grateful to the Research Institute for Subtropical Agriculture and Biotechnology and Sustainable Agriculture Research Institute (SARI) at Jeju National University for providing experimental facilities.

**Conflicts of Interest:** The authors declare no conflict of interest.

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
