# Peer review of "Evaluation of Fermented Extracts of Aloe vera Processing Byproducts as Potential Functional Ingredients"

_fermentation, doi:10.3390/fermentation7040269_

Round 1
Reviewer 1 Report
in vitro evaluation of fermented extracts of Aloe Vera by-products is presented systematically in this paper, and thus it could be considered as a scientific novelty. This is an interesting essay to study the potential of Aloe Vera as a functional/active ingredient in cosmetics and even in nutricosmetics or nutraceuticals, but especially regarding the appropriate efficacy evaluation for cosmetic products, however in the present form of the manuscript there can be found some technical weaknesses of approach, instrumentation, analysis and/or interpretation.
I permit to give some suggestions, which could enhance the quality of the manuscript:
Introduction
Line 40- "skin disease" or skin diseases?- I suggest to give some examples, many bibliographic studies report a benefic effect by using Aloe sp. for the improvement of different skin diseases (for e.g.
Treatment of Skin Disorders with Aloe Materials, Curr Pharm Des., 2019;25(20):2208-2240. doi: 10.2174/1381612825666190703154244.
Miroddi, M.; Navarra, M.; Calapai, F.; Mancari, F.; Giofrè, S.V.; Gangemi, S.; Calapai, G. Review of clinical pharmacology of Aloe vera L. in the treatment of psoriasis. Phyther. Res. 2015, 29, 648–655. doi: 10.1002/ptr.5316.)
Also, valuable articles report data regarding adverse effects of Aloe sp. in cosmetics and topical products, aspect that in my opinion should be mentioned in the manuscript (for e.g.
Final Report on the Safety Assessment of Aloe andongensis Extract, Aloe andongensis Leaf Juice, Aloe arborescens Leaf Extract, Aloe arborescens Leaf Juice, Aloe arborescens Leaf Protoplasts, Aloe barbadensis Flower Extract, Aloe barbadensis Leaf, Aloe Bar. Int. J. Toxicol. 2007, 26, 1–50. doi: 10.1080/10915810701351186.
Guo, X.; Mei, N. Aloe vera: A review of toxicity and adverse clinical effects. J. Environ. Sci. Heal. Part C Environ. Carcinog. Ecotoxicol. Rev. 2016, 34, 77–96. doi: 10.1080/10590501.2016.1166826.)
Materials and Methods/Results and Discussion
Regarding efficacy evaluation, the current legislation of cosmetics underlines "where justified by the nature or the effect of the cosmetic product, proof of the effect claimed for the cosmetic product" is mandatory (REGULATION (EC) No 1223/2009 OF THE EUROPEAN PARLIAMENT AND OF THE COUNCIL of 30 November 2009 on cosmetic products, Article 11; https://www.cosmeticsandtoiletries.com/regulatory/region/asia/Korean-Cosmetic-Regulations-199871081.html). Indeed, the authors describe the performed in vitro studies and different activities of Aloe vera by-products, but instrumental evaluation of different biophysical parameters like moisturizing (coneomery/TEWL evaluation), depigmentant/skin-whitening (colorimetric evaluation) and or anti-aging effect (cutomery, for e.g.) should be performed for the evidence of the efficacy and potential as an active ingredient.
Line 251-252- "Excessive melanin causes skin pigmentation, such as spots and freckles"- hyperpigmentation, melasma?
Conclusions
Like mentioned before, a clear evidence and evaluation is necessary to demonstrate and sustain the activity and application of Aloe by-products in cosmetics.
Kind regards,
the Reviewer.
Reviewer 2 Report
The title should be shorter
In the Figure 1/2/4/5 should appear the significative differences between columns.
Round 2
Reviewer 1 Report
Dear authors,
thank you for providing the suggested corrections of your manuscript.
Please take into consideration my minor observations:
Line 260-261: "Excessive melanin causes skin pigmentation, such as hyperpigmentation and melasma"- eventually "Excessive melanin can cause hyperpigmentation and melasma"- please avoid repetitive terms
Line 311: "Drug Administ-ration (KFDA)"- please correct "Administration"
Regarding efficacy studies of the active ingredient, I can understand it is not possible to conduct them immediately and of course they suppose considerable financial effort, but legal aspects at the moment consider these evaluation mandatory, aspect that please take into consideration in the future. Thank you for mentioning at least biophysical parameters evaluations regarding cosmetics efficacy in Conclusions.
I wish you all the best!
Kind regards,
the Reviewer.
Author Response
We are appreciating for your minor observations.
Line 260-261 : we changed as "Excessive melanin can cause hyperpigmentation and melasma"
Line 311 : we changed as "Drug Administration (KFDA)"
Kind Regards
Authors